# Benchmarking Community Disaster Resilience in Nepal

**DOI:** 10.3390/ijerph17061985

**Published:** 2020-03-18

**Authors:** Sanam K. Aksha, Christopher T. Emrich

**Affiliations:** School of Public Administration and National Center for Integrated Coastal Research, University of Central Florida, Orlando, FL 32816, USA; christopher.emrich@ucf.edu

**Keywords:** disaster, community resilience, Nepal, Himalaya, natural hazards, Community Disaster Resilience Index (CDRI)

## Abstract

Building disaster resilience is a stated goal of disaster risk reduction programs. Recent research emphasizes a need for a greater understanding of community disaster response and recovery capacity so that communities can absorb shocks and withstand severe conditions and progress through the recovery period more efficiently. Nepal, which is prone to a multitude of hazards and having recently experienced a large earthquake in 2015, provides a unique opportunity for exploring disaster resilience in the developing world context. To date, no study investigating community disaster resilience across the entire country of Nepal exists. This study quantifies disaster resilience at Nepal’s village level, primarily using census data. Guided by the Disaster Resilience of Place (DROP) model, 22 variables were selected as indicators of social, economic, community, infrastructure, and environmental resilience. Community resilience was assessed for 3971 village development communities (VDCs) and municipalities while using a principal component analysis. Additionally, a cluster analysis was performed to distinguish spatial patterns of resilience. Analyses reveal differential community disaster resilience across the country. Communities in the capital city Kathmandu and in the western and far western Hill are relatively resilient. While the entire Tarai region, which holds the greatest proportion of Nepal’s population, exhibits relatively low levels of resilience when compared to the rest of the county. The results from this analysis provide empirical evidence with the potential to help decision-makers in the allocation of scarce resources to increase resilience at the local level.

## 1. Introduction

Global climatic and environmental changes have escalated both the frequency of and losses from disasters in recent years [1,2,3]. Consequently, forced displacements due to disasters, together with economic and social marginalization, are challenging communities’ capacity to build resilience against shocks and stressors, such as those that are associated with disaster events. Thus, understanding a community’s ability to respond and recover from disasters and enhancing the overall capacity to build safer communities has been a major priority of many disaster risk reduction (DRR) programs and policies [4]. In this case, resilient communities are defined as “societies which are structurally organized to minimize the effects of disasters, and, at the same time, have the ability to recover quickly by restoring the socioeconomic vitality of the community” [5].

In recent decades, researchers and practitioners have become increasingly interested in measuring a community’s degree of resilience as a starting point for developing strategies and taking actions toward the effective implementation of DRR programs and policies that aimed at building community disaster resilience [6,7]. Likewise, international development organizations have applied enormous effort and resources toward building resilient communities so that shocks and stresses can more easily be absorbed during adverse conditions, and communities can bounce back better and more quickly move toward resilience. Such abundant resources being devoted to these programs should put communities in a better position to build adaptively/sustainably into the future. The Sendai Framework for Disaster Risk Reduction (SFDRR) clearly states that building disaster-resilient communities is the major goal for 2015–2030. Out of its four priorities of action, ‘investing in disaster risk reduction for resilience’ is one of the main thrusts of the programs. Building from the Hyogo Framework for the Action (2005–2015) achievements, SFDRR calls for the higher investment in disaster risk reduction programs to build more resilient communities [8]. However, an evaluation of progress toward community resilience requires an initial baseline understanding of community resilience. How can one measure “more” resilient communities without first understanding where a community is situated on a resilience trajectory? In Nepal, no such understanding of resilience exists. Analyzing and benchmarking community resilience based on available relevant community characteristics and variables can be utilized to address this gap.

Resilience measurement, primarily operationalized through metrics and framework development, provide baseline indices that offer rich insight into a community’s disaster resilience [6,9]. Based on specific community characteristics depicting broad concepts of resilience, pre-event conditions of a community can be assessed, evaluated, and compared across space and (given enough data points) across time. Baseline resilience measurements can serve as a focal tool for formulating effective programs and policies prior to the hazard event occurrence. However, indicator selection requires careful consideration, ensuring a reliable reflection of the study area characteristics [10,11]. While various methods and measures have been used to examine and estimate community disaster resilience in the developed countries, very few models have been applied toward understanding disaster resilience in developing nations.

Being guided by the Disaster Resilience of Place (DROP) model, this study examines the following research questions in developing community disaster resilience index for Nepal: How does community resilience manifest itself across Nepal?Do clusters of higher and lower community resilience exist across Nepal?

Nepal, in particular, is exposed to a multitude of natural hazards and experiences hazardous events on a regular basis [12]. In 2015, a 7.8 magnitude of earthquake took about 9000 lives and damaged over two million houses, including many critical infrastructure elements, such as hospitals, roads, and bridges [13]. Moreover, monsoonal floods and landslides, regularly claiming hundreds of lives, damaging buildings, and critical infrastructure, and impeding economic activities annually, impact the country. Table 1 exhibits the major disasters and their impacts (deaths, injuries, and economic losses) that occurred between 2008 and 2019 in Nepal. Although the country is battered by various disasters each year, there remains a paucity of research that is focused on community resilience across the nation. This study identifies resilience indicators and develops a community disaster resilience index (CDRI) for Nepal. Such measures provide a baseline set of community-level resilience information that was linked with pre-disaster conditions at the village level. Further, this study explores the spatial distribution of resilience, patterns of high/low resilience scores, and the clustering of hot/cold resilience across the country.

## 2. Concept of Resilience and Its Measurement

### 2.1. Defining Resilience

Resilience is a complex and multifaceted term that is continuously evolving in meaning and gaining prominence across many disciplines. Generally, the term describes how a system can overcome stresses and shocks. Originally, the word resilience comes from Latin root *resi-lire*, which means to spring back [17], and the first dictionary definition can be traced back to the 17th century, where it is used in dual meaning: to rebound and to go back on one’s word [18]. However, the concept became very popular when Holling (1973) described the resilience concept as “a measure of the persistence of systems and of their ability to absorb change and disturbance and still maintain the same relationships between populations or state variables” [19]. Today, the concept is omnipresent and it has influenced various disciplines and different fields, such as ecology, engineering, geography, and psychology, and it has been used in academic, political, and policy discourses [5]. Nonetheless, one general conceptual consensus among many disciplines is resilience is the “ability to prepare and plan for, absorb, recover from, or more successfully adapt to actual or potential adverse events” [20]. 

Despite more than five decades of collective conceptual and research and experience, resilience is still a contested term that has mainly arisen from different epistemological orientations and subsequent methodological practices [9,10,21]. Subsequently, resilience does not have an internationally agreed-upon definition. A literature survey found 60 different definitions that are used between 1996 and 2013, which demonstrates the popularity of the term as well as disagreement among scholars over the use of the term. The United Nations Office for Disaster Risk Reduction (UNISDR), among the foremost international agencies that are tasked with disaster preparedness, defines resilience as: “the ability of a system, community or society exposed to hazards to resist, absorb, accommodate to and recover from the effects of a hazard in a timely and efficient manner, including through the preservation and restoration of its essential basic structures and functions”. Despite its conceptual ambiguity, it is a useful concept and it has the potential to offer a more systematic and cross-cutting approach to disaster risk reduction, climate change adaptation, and the humanitarian sector [22,23].

### 2.2. Measuring Disaster Resilience

The measurement of a community’s natural hazard and disaster response and recovery has been an emphasis on recent disaster research [4,9,24,25]. Converting a conceptual framework of resilience into an empirical model has been impeded by continued disagreements across disciplines on simply defining resilience among other challenges, including whether it is an outcome or process, the specific focus of the resilience (social, economic, etc.), and target of the research [26]. Nonetheless, researchers and practitioners have proposed different methodologies and frameworks for measuring disaster resilience employing qualitative and quantitative approaches at the community, regional, and national levels [11,27,28]. These studies primarily focus on establishing baseline resilience conditions for use in monitoring the progress of communities across space and time.

A review of employed approaches can be broadly classified into two categories: qualitative and quantitative. Qualitative approaches are primarily adopted in a small-scale study to understand underlying vulnerability and community capacity, and determine how a specific community prioritizes their risks and perspectives on resilience factors [20,29,30]. Unlike methods, such as in-depth interviews, focus group discussion, life stories, and/or observations, which are commonly used in qualitative approaches to explore community resilience [28], quantitative approaches attempt to make resilience comparable between geographic location, entities (e.g., households, communities, businesses), and across time predominantly employing indicator-based measurements [27]. Utilizing numerical information to empirically measure or characterize places, quantitative measures often result in indices (quantifying variables of selected characteristics), scorecards (evaluating progress toward a goal), and tools (modeling simplified representation of systems) that are built from secondary data or survey results [4,28,29].

Various metrics have been developed in recent decades in order to evaluate different types and numerous interpretations of resilience. Mavhura and Manyena [10] listed 43 tools that were developed by academia and disaster practitioners under various categories, such as toolkits, models, indexes, scorecards, and policy guides to quantify resilience. However, resilience measures have primarily been a developed world phenomenon. Indeed, in the United States, for example, 27 different disaster resilience assessment approaches were identified to assess the landscape of disaster resilience indicators [9]. Conversely, developing countries have primarily focused on assessments of coping capacity to uncertainty and shocks. In developing countries, vulnerable populations are regularly exposed to the severity of natural hazards and disasters requiring less focus on resilience and a heavier emphasis on more immediate response and coping capacities [10]. Current resilience indices, both US-based and international, have been systematically reviewed and summarized by multiple scholars, including Asadzadeh, Kötter [31], Cutter [9], and Patel, Rogers [32]. Table 2 presents these indices, methodological approaches, study areas, domains, and indicators. Among the most prominent indices is the Baseline Resilience Index for Communities (BRIC) [6,10,11], a county-level index derived through an in-depth literature review linking community-specific disaster resilience measures to available datasets in the United States context. We chose to replicate BRIC methods because of the geospatial context that is inherent in the model as well as the latitude in indicator selection linking theory to the best measures that are available in Nepal.

The theoretical framework provided by the Disaster Resilience of Place (DROP) model guides this research [7,26,42]. Although the model was initially designed for the United States context, its foundational connections to disaster resilience as a whole make it transferable to other country contexts. DROP has been adopted and adapted in several other countries, such as Australia [11], Norway [28], Thailand [6], and Zimbabwe [10]. The model assumes that natural systems interact with social and built environment systems to produce antecedent conditions that contain both inherent vulnerabilities and inherent resilience [42]. While inherent vulnerabilities determine the ability of a population to prepare for, respond to, and recovery from, disasters; inherent resilience identifies community characteristics that might accentuate or attenuate its capacity to prepare for, respond to, and recover from, and mitigate environmental hazards that are assumed to be in place prior to the onset of a hazardous event [7]. The DROP model presents a comprehensive framework for evaluating the overall community disaster resilience and provides an opportunity to pinpoint those specific components that may enhance or reduce disaster resilience. Furthermore, the application of the DROP model permits an understanding of geographic patterns of disaster resilience and enables spatial comparison among communities.

Disaster resilience and community disaster resilience studies are at the nascent stages in Nepal. Existing studies are either focused in a specific geographic location [43,44] or hazard types, such as earthquake [45,46,47], flood [48], and landslide [49]. In the prior cases, mostly international development agencies have initiated and funded such efforts, being published in gray literature, and often driven by project requirements, whereas, in the latter case, the 2015 Nepal earthquake played a pivotal role for many researchers to delve into this topic. However, there is no overarching research that provides a comprehensive and comparable picture of community disaster resilience across the country.

## 3. Materials and Methods 

### 3.1. Study Area

Nepal is situated in the central Himalayan region and it covers approximately one-third of its area. Nepal is surrounded by India in the east, west, and south, and bordered by China in the north. The country is characterized by diverse topography, climate, culture, language, and religion; thus, depicting spatial heterogeneity within a short areal distance (north to south average areal distance is about 200 km). The total area of the country is 147, 141 square kilometers, and, according to the recent census, the total population of the country is about 26.5 million [50]. 

From the ecological perspective, Nepal is primarily divided into three eco-regions: Mountain, Hill, and Tarai (Figure 1). Within about 200 km of Nepal’s North-South extent, the altitude differs from 57 m above sea level to Mt. Everest (the world’s highest peak) at 8848 m. Nepal’s mountain region consists of about 19% of total land and it is primarily characterized by the presence of snow-capped mountains. The altitude typically varies from about 4000 m. to 8,8848 m. and it consists of glaciers, glacial lakes, and tundra environments. The Hill region dominates Nepal’s topography, which comprises approximately 64% of the total area. Hill altitude ranges from about 600 m. to 4000 m. above sea level. The area is significantly populated and relies on subsistence agriculture. The major cities, such as the capital city Kathmandu, Pokhara, and Surkhet, are situated in the Hill region, and sub-tropical to temperate climate dominates this region. Tarai is the southernmost region of the country, which represents approximately 17% of the total land. The Tarai is densely populated, as about 48% of the total population is living within this region. This region is considered as the ‘grain basket’ of the country because of high agricultural productivity and relatively flat terrain. 

In 2015, Nepal adopted a new constitution and federal governance model with three levels: the federation, the province, and local bodies. Currently, there are seven provinces and 753 *Palikas* (six metropolises, 11 sub-metropolises, 276 municipalities, and 460 *gaunpalikas*). Previously, Nepal was divided into five development regions, 14 zones, 75 districts, 53 municipalities, and 3918 village development committees (VDCs). We adopted the previous administrative units (3918 VDCs and 53 municipalities) as our unit of analysis, because the current local government bodies are essentially formed, appending numerous VDCs and municipalities. Thus, previous administrative units are finer in scale and they provide a more comprehensive picture of disaster resilience across the nation.

The diverse topography and distinct social and demographic distribution intersect with geophysical and hydrometeorological processes, which makes Nepal a hazardous landscape that is exposed to several natural hazards. Flood and landslides are the most frequent hazards, together contributing to two-thirds of disaster mortality [51] and causing significant economic and infrastructure damages every year. South Asian Monsoon, which occurs between June to September, supports subsistence agriculture, but it generates disastrous wet landslides, mudslides, and floods annually.

Additionally, Nepal is impacted by low frequency/high magnitude disasters events, such as earthquakes. In a recent example, the 2015 Gorkha earthquake (7.8 magnitudes in Richter scale) resulted in about 9000 fatalities damaged hundreds of thousands of buildings, which were mostly rural traditional mud houses, and had USD 7 billion economic impacts [37]. Fourteen districts (Bhaktapur, Dhading, Dolakha, Gorkha, Kathmandu, Kavre, Lalitpur, Nuwakot, Okhaldhunga, Ramechhap, Rasuwa, Sindhupalchowk, and Sindhuli) were worst hit and they were declared ‘crisis-hit’ by the government to prioritize rescue and relief operations [37]. This disastrous event exposed inadequate disaster preparedness at the national, regional, and local level governments, and it has demonstrated the importance of community disaster resilience research to understand and eventually improve ex-ante, during, and ex-post disaster response and recovery. 

### 3.2. Selection of Variables 

Based on the literature and guided by the DROP model [7,26,42], 22 different social, economic, infrastructure, environmental, and community resilience indicators for 3,971 VDCs and municipalities are used in this study. Table 3 describes indicator and variable details, data sources, and impact on resilience. The DROP model [7] used 49 variables in six subdomains of the resilience concept to construct a community resilience index at the county level across the country. Since the model is designed for the United States context, it is evident that many variables are different in the international context as a result of social, cultural, political, economic, and geographic differences. Hence, the variables were carefully chosen, focusing on two things: 1) suitability of the variables in the Nepali context. For instance, we added Dalit Population in the model, as Dalit represents the lowest strata in the caste system of Nepal and it is characterized by a lower level of resilience [52,53]. Additionally, we added Absentee Population to reflect prevalent male outmigration mainly to the Gulf countries (Qatar, Saudi Arabia, United Arab Emirates), Malaysia, and South Korea [54,55], and 2) the availability of the data at the village level. Many variables that are used in the DROP model, such as variables related to institutional resilience (mitigation spending, local disaster training, flood insurance, crop insurance), are not available at the village level in Nepal. 

Census reports serve as the primary data source with a few variables that were collected from other relevant studies. Census data were used, because these are the most robust datasets in the country. A desire to elucidate the many different resilience functions at play across Nepal required the application of the PCA model rather than an additive model used in Cutter et al.’s [7,26,42] seminal resilience measurement study.

### 3.3. Methods

A total of 22 variables were selected for the construction of the community disaster resilience index (see Table 3). The raw variables were transformed while using percentage, per capita, or density functions as appropriate for the type of data/variable being used. Since Nepali communities are comprised of varying sizes, population densities, characteristics, a transformation of the variables permits comparability across the unit of analysis. The transformed variables were then standardized while using z-score standardization in SPSS version 25 (IBM Corp, New York, USA) [57]. The Bartlett sphericity test (with p < 0.05) and the sampling adequacy measure Kaiser-Meyer-Olkim (KMO) (with selection criterion of values between 0.7 and 1) was used to determine whether the chosen variables were adequate for the principal component analysis while following the methodological approach that was used in various index construction methods [6,10,52,58]. 

Next, a principal component analysis applying a varimax rotation and Keiser criterion was employed as the extraction method for components. Varimax rotation was chosen to minimize the number of resulting components and maximize the sum of the variances they represent. The Kaiser criterion was applied to extract the number of factors while using eigenvalues greater than 1 as a cutoff for inclusion. Each component theme was generated based on the characteristics of its variables, specifically its dominant variable. All of the component scores were summed to construct the community disaster resilience index (CDRI) for each village and municipalities (3918 village development committees and 53 municipalities). The equal and additive approach was used in the absence of empirical and justifiable evidence for weighting components differently, as has been exercised in similar studies [6,52,58]. The CDRI scores of each spatial unit were mapped while using ArcMap version 10.5 to visualize the most and the least resilient villages in Nepal based on the standard deviation from the mean value.

We used the Getis−Ord G* test of spatial autocorrelation to investigate CDRI score clustering or randomness across space in statistical terms. The Getis−Ord Statistics method is an efficient method for expressing the spatial relationship between different samples [59]. In this study, it was used to depict the high-value and low-value clusters of CDRI. A village will have a high G*i** value and be surrounded by other villages with high G*i** values as well to be statistically significant hot spot of CDRI; on the contrary, to be a statistically significant cold spot of CDRI, a village will have a low G*i** value and be surrounded by other villages with low G*i** values.

## 4. Results

### 4.1. Components of CDRI

The KMO and Barlett’s test prior to the principal component analysis yielded a KMO value of 0.813, which indicated that the variables included in the model are ‘meritorious’ for conducting the PCA [60]. Using varimax rotation and Kaiser normalization, the PCA resulted in six components explaining 69.05 % of the total variance of the data. Major themes were identified for each component based on the loadings of the variables. These components were subsequently labeled as Infrastructure, Economic-Social, Community Capital, Environmental, Caste, and Migration. Table 4 presents the factors, dominant variables, and factor loading. A brief explanation of these factors is presented below.

The first component, labeled ‘Infrastructure’, explains 26.05% of the variance of the data. Variables, such as percentage of RCC buildings (those reinforced cement slab structures), percentage of households with an internet connection, and percentage of the population not employed in a primary occupation, such as farming, are the major contributor to this component, while the percentage of owner-occupied housing units, percentage of the population born in the same place, and percentage of the population without school education are major detractors in this component. Based on the loadings of variables, the second component, labeled ‘Economic-Social’, explains 18.72% of the variance of the data. Economic indicators, such as percentage of the labor force employed and percentage of female employment, social indicators, such as the percentage of the population who speaks the Nepali language are a major contributor to this component. Transportation and non-special need variables are the major detractors in the second component. The third component, labeled ‘Community Capital’, explains 8.94% of the variance of the data. Absentee population and female-headed households’ variables are the major contributors, whereas pre-retirement age is a major detractor in this component. The fourth component, labeled ‘Environmental’, explaining 5.56% of the variance, has two environmental variables (elevation and pervious surfaces) loading positively. The fifth component, labeled as ‘Caste’, has one variable (Dalit population) and it explains 5.01% of the variance of the data. The sixth component, labeled ‘Migration, loaded with the migrated population variable, explains 4.77% of the variance of the data.

### 4.2. Geographic Distribution of CDRI Components

Figure 2 displays the geographic distribution of each principal component. For each component, a mean and standard deviation (SD) of the scores was calculated. Based on the SD values, the resilience component scores were grouped into five classes from least (< -1.5SD) to the most resilient (>1.5SD). Using ArcMap version 10.5, the values of each component are plotted to visualize the spatial distribution of community disaster resilience across Nepal (Figure 2). The resulting maps highlight the diverse geographic location of villages with significantly high and low resilience scores for different resilience components. The Infrastructure map shows that high resilience villages are distributed in the capital city, Kathmandu, as well as in the eastern Tarai districts. Likewise, high resilience areas are found in the Hill and Mountain villages for the Economic-Social and Environmental components. The majority of highly resilient villages that were identified by Community Capital and Caste components are distributed in western Hills. The Migration component is somewhat distributed evenly throughout the country. 

### 4.3. Spatial Distribution of CDRI Scores

The Community Disaster Resilience Index (CDRI) scores were calculated by summing all six principal components. Next, the mean and standard deviation (SD) of the CDRI scores were calculated. Based on the SD values, the resilience scores were grouped into five classes from least (< -1.5SD) to the most (>1.5SD) resilient. Using ArcMap version 10.5, CDRI scores were mapped (Figure 3). The overall CDRI map depicts that the most resilient villages are primarily distributed in the districts of western and far western Hill region (Table 5). Few villages in the eastern and western Mountain region are also depicted as being highly resilient places. The capital city, Kathmandu, is also categorized as highly resilient areas. However, nearly the entire Tarai region is identified as among the least resilient places. The majority of Hill and Mountain regions are characterized as medium to high resilient region.

### 4.4. Spatial Agglomeration of CDRI Scores

A cluster map of the overall CDRI scores was prepared while using the Getis−Ord G*i** Statistics tool in ArcMap 10.5, depicting the spatial agglomeration characteristics of CDRI across the country. The resulting “Hot spots” indicate that a village had high CDRI and it was also surrounded by villages with high CDRI, and “cold spots” mean that a village had low CDRI and it was also surrounded by villages with low CDRI. The Getis-Ord G*i** Statistics tool generates results of a hot spot and cold spot in 99%, 95%, and 90% significance level. We only included 95% and 99% as clusters of high and low resilience. All other villages with < 95% significance were classified as ‘Not significant’ (Figure 4). Figure 4 shows the least resilient villages are clustered in the central and western Tarai, and the villages with high CDRI scores are clustered in western and far western Hill regions. Districts, such as Arghakhachi, Baglung, Gulmi, Kaski, Lamjung, Myagdi, Parbat, Pyuthan, Syangja, and Tanahun, and capital city Kathmandu, have clusters of highly resilient communities, whereas districts, such as Bara, Parsa, Rautahat, and Sarlahi, have clusters of least resilient communities. Few communities in the eastern and western Mountain have a cluster of highly resilient villages.

## 5. Discussion

This study constructed a community disaster resilience index (CDRI) to benchmark those baseline conditions across Nepal. The results can provide a measure from which to monitor changes in disaster resilience over time. In addition, this paper mapped the geographical distribution of community disaster resilience at the local level while using indicators relevant to Nepal’s distinct social and physical landscape. Being guided by the DROP model, we identified six components that contribute to disaster resilience in Nepal. Our study provides a nation-wide comparison at the village level while using the most complete and comprehensive datasets available.

The spatial distribution of CDRI scores (Figure 3 and Figure 4, and Table 5) shows that the majority of the Tarai region falls under low and medium to low resilient category. It is important to note that the Tarai region accommodates approximately half of the entire population. Perhaps not so coincidentally, most of these locations are also socially vulnerable [41]. Further, a recent increase in rural (Hill) to urban (Tarai) migration exposes a significant number of vulnerable populations towards serial and sporadic natural hazards, thus exacerbating their situation [54]. In recent years, during monsoon season, the Tarai region has experienced recurring riverine and flash flood events and inundation along the Nepal India border. For instance, in the summer of 2015, 2016, 2017, and 2018, each monsoonal flood devastated this region claiming hundreds of lives, damaging vital transportation networks, and destroying critical infrastructures [61,62]. The probable intersection of growing hazardous events, increasing social vulnerability, and variable resilience weakens the capacity of ‘at risk’ communities to respond and recover from any disasters. 

Geographic distribution of individual resilience components (Figure 2) categorizes different villages with high and low scores. The first component, ‘Infrastructure’, reflects the general trend of urbanization focused in the capital city and along the Tarai region, although a few exceptions are observed in the Mountain region. Likewise, the ‘Environmental’ component shows that capital city Kathmandu is listed in the high category; however, it has the lowest pervious surfaces and it located at a higher elevation than the entire Tarai region. Further, our results reveal that very few “worst hit” communities of the 2015 mega earthquake in eight districts (Dhading, Dolakha, Gorkha, Kavre, Nuwakot, Rasuwa, and Sindhupalchowk) of the central region of the country are categorized as being highly resilient communities. This finding reminds us that measuring community resilience can be scale and context-dependent, and such findings warrant further studies to see the relationship between disaster outcomes and level of community resilience [47]. 

A very interesting regional difference is observed between the eastern and western Hill regions given the similar kind of exposure to geophysical and hydro-meteorological hazards (Figure 3 and Figure 4). Our spatial clustering map (Figure 4) identifies a few clusters of high resilient villages in far western Hill districts. These areas have the highest poverty rate in the country (46% living below poverty line), facing chronic food insecurity, and they are facing acute and chronic shocks and stresses, such as flood, landslide, hailstorm, drought, forest fire, low water availability, and soil degradation [43]), and they were categorized as high social vulnerability region [52]. Likewise, few western Hill districts (Arghakhanchi, Baglung, Gulmi, and Pyuthan) were classified under the highly vulnerable category in a companion study [52]. On the contrary, these districts are clustered with high CDRI scores in the current study (Figure 4). These Hill districts, including Kaski, Parbat, Syangja, and Tanahu, have higher development patterns [63] following Kathmandu valley and higher water productivity than other districts in the country [64]. Few clusters in the eastern Mountain region could be attributed to well-developed tourism infrastructure, which has high impacts on our CDRI variables [65,66]. Our study shows, in comparison to Aksha, Juran [52], that resilience and vulnerability are not opposing concepts to each other, but rather they do overlap to some extent [5,22]. We employed a bivariate mapping technique to visualize the relationship between social vulnerability and community resilience (Figure 5). We mapped five categories (High, High-Medium, Medium, Medium-Low, and Low) of social vulnerability [52] and community resilience at the local level, and found that the majority of Low and Medium-Low resilience categories, and High and High-Medium vulnerability, are distributed in the Tarai region (Figure 5). Similarly, High resilience and High vulnerability, and Low resilience and Low vulnerability are primarily dispersed in the Hill regions of the country. 

We compared distributions of percent villages and percent population in each CDRI category in each district to further elucidate the resilience concept and tie results to specific places on the ground, and found some interesting results (Table 6). In particular, Table 6 shows that urban districts, such as Bhaktapur, Chitwan, Kathmandu, and Lalitpur, and Tarai district, such as Rupandehi, have significantly higher population percentages that reside in high resilience areas than percent villages exhibiting high resilience. For instance, Kathmandu district comprises 54.24% of villages under high resilient categories, while it consists of 92.52% of the total population under the same category. Similarly, Rupandehi has only 1.41% of VDCs under high resilient categories, but these VDCs hold 13.55% of the population. In effect, this means that, although there are not large numbers of villages exhibiting high resilience within a district, those that do have higher resilience have more people residing in them. It shows the uneven distribution of population within a geographic boundary and calls for further investigation. 

Likewise, the population percentage is always lower than the percentage of VDCs in low resilient categories. Few districts, such as Kailali, Lalitpur, Mustang, and Udayapur, have a significant difference in the medium category of CDRI. This comparison (Table 6) provides a unique perspective to researchers and practitioners to compare the geographic distribution of community disaster resilience with population and it should be helpful for planners and decision-makers who are routinely looking for the empirical measure when making critical choices regarding who is at risk, vulnerable, and in need of assistance before, during, or after disasters.

Building resilience is a long-term disaster management strategy that requires adequate investment in mitigation, risk reduction, and risk management activities through relevant programs and policies. Building resilient communities has not been a priority of the government although Nepal is exposed to various natural hazards and the disaster outcomes are escalating every year. However, the Nepali government enacted a new disaster management act, the Disaster Risk Reduction, and Management (DRRM) Act, in 2017, which replaced a four-decade-old disaster act, Natural Calamity (Relief) Act 1982. However, Nepal is still struggling to form a functional coordinating agency envisioned in the new act. Moreover, the international development agencies played a positive role to forward disaster risk reduction and resilience in the country, despite the absence of coordinating agency, legislative tools, and political impasse. However, the international development agencies also lack a comprehensive view of disaster resilience, and it is often used as an operational tool for influencing the Government of Nepal [67]. Additionally, disaster risk reduction activities are primarily focused on a single hazard type. In this way, hazards are perceived as isolated natural processes, and their cascading effects are completely neglected [68]. Consequently, preparedness and mitigation efforts are deficient in terms of their ability to minimize future impacts from all-to-often compounded events.

Although quantitative approaches are widely used and offer a more systematic and reliable way to measure various dimensions of resilience, the approach has several drawbacks. Quantitative approaches fail to capture less tangible elements of resilience, like social capital or power relations, are often data-intensive and data-driven, and may include researcher’s bias during indicator selection—which is itself a very context-specific as a resilience factor in one community might not be a resilience factor in another community [69]. Qualitative approaches can complement quantitative frameworks by further including the intangible factors of resilience, such as social cohesion, risk perception, and power relations [29]. 

The proposed community disaster resilience index provides an effective tool for identifying community natural hazard resilience individually and in clusters and evaluate, in the context of disaster prevention, emergency rescue, and post-disaster recovery and reconstruction. However, several limitations and deficiencies in this study remain and they should be improved into the future. First, our work could not include variables that were related to institutional resilience, which significantly contributes to overall resilience. For example, disaster mitigation spending, flood insurance coverage, crop insurance coverage, and local disaster training are missing. Unlike the US context, such data is unavailable in Nepal. Second, after normalization of data, the final score is a relative value leaving users without an absolute measurement of community resilience for a given location. Although such relative estimation provides easily understood comparisons between places and it is useful for benchmarking progress over time and across space, it could under- or overestimate community resilience at a particular location. Third, principal component analysis is an efficient way to identify dominant variables in the CDRI, but it cannot explain the dynamic and overarching nature of community resilience [59]. It is difficult to quantify resilience in many instances because of the qualitative nature of many resilience indicators. Fourth, although census data are the most complete and comprehensive datasets available in Nepal, they lack, including many variables that are relevant to benchmark and measure progress towards disaster resilience and sustainable development in the country. Additionally, it is worth noting that data gathering in high Mountain and Hill areas (refer Figure 1) is challenging, as these areas are not adequately connected to national transportation networks and pose communication difficulty in many communities, since they may not speak/understand the Nepali language.

## 6. Conclusions

Climatic and environmental change will continue to increase the stresses and threats to communities globally. Thus, reactive approaches to disasters that focus on recovery and reconstruction are a costly and inefficient means to deal with the challenges that are posed by them. Building resilient communities with higher response capacity that will more rapidly recover from future disasters is a meaningful way of investing resources. While place-based indicators of community resilience only provide a single comprehensive measurement of community resilience benchmarking current conditions, such a measure provides a vital first step in moving positively along a resilience continuum. How can improvements in community resilience be made, and how can places become “more resilient” without an initial understanding of community-level resilience as a baseline measure? Only with an understanding of where our communities are currently situated in terms of disaster resilience can future changes in resilience be measured and monitored over time and across space. Using the aftermath of the 2015 Gorkha earthquake as an example, the Nepal government has invested significant resources on recovery and reconstruction with a majority of this investment on rebuilding infrastructure. While Nepali government efforts are commendable in terms of post-earthquake recovery and reconstruction, we highlight the need for equal emphasis on building a more holistic community disaster resilience strategy that is focused on all aspects of a community’s long term ability to respond and rebound from shocks and stresses.

Our study provides a geospatial and visual depiction of community disaster resilience across Nepal. The distribution and pattern of Nepal’s CDRI could inform policymaking, resource allocation, and disaster management among government officials and non-governmental organizations that are focused on improving future outcomes across the nation. As a majority of the data of this study are based on census 2011 data, and it can be periodically revised to monitor temporal changes and measure the impacts of disaster programs and policies, Nepal’s path forward should include building programs and policies aimed at increasing community resilience in concert with constructing more disaster-resistant infrastructure. Stated Nepalese and Sendai framework goals can only be realized once communities become a more significant piece of the resilience equation. In this vein, future research in community disaster resilience should focus attention on the differences in disaster resilience in rural and urban areas and the role of pre-disaster baseline conditions on the community’s capacities on recovery and reconstruction after the 2015 Gorkha earthquake. 

## Figures and Tables

**Figure 1 ijerph-17-01985-f001:**
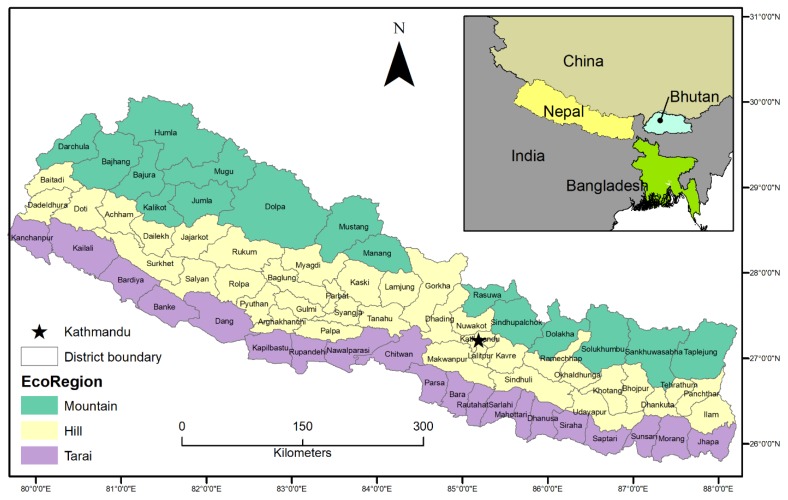
Administrative map of Nepal.

**Figure 2 ijerph-17-01985-f002:**
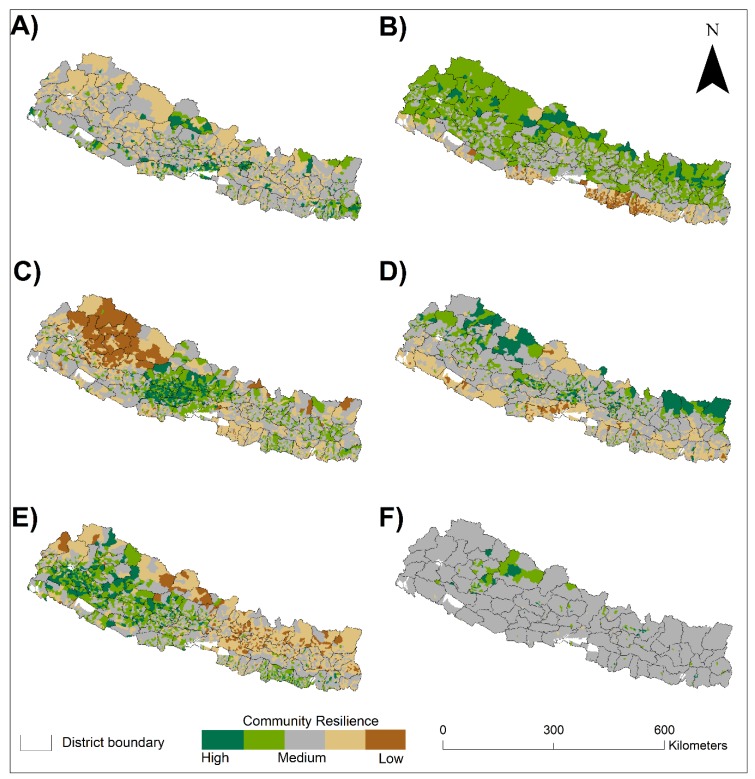
Geographic distribution of community disaster resilience index (CDRI) components: (**A**) Infrastructure, (**B**) Economic-Social, (**C**) Community Capital, (**D**) Environmental, (**E**) Caste, and (**F**) Migration.

**Figure 3 ijerph-17-01985-f003:**
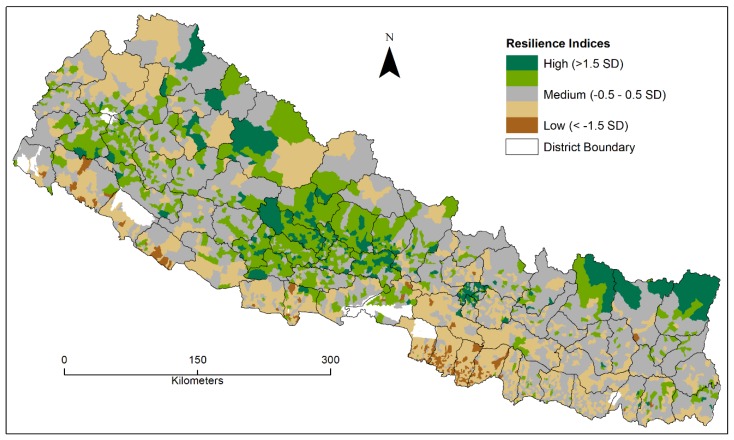
Spatial distribution of CDRI scores.

**Figure 4 ijerph-17-01985-f004:**
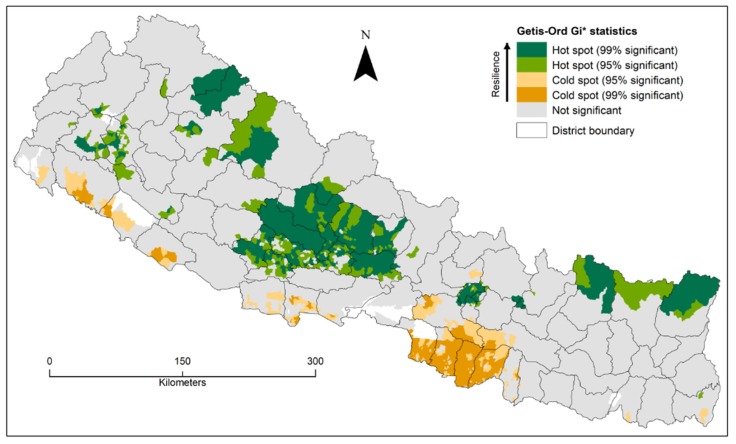
Spatial agglomeration of CDRI.

**Figure 5 ijerph-17-01985-f005:**
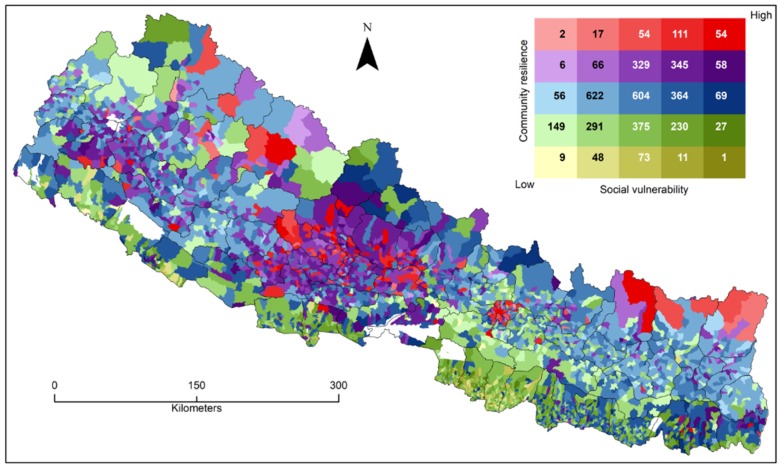
Bivariate map (5x5 categories) depicting social vulnerability (SoVI) and community disaster resilience (CDRI) at the village level. Numbers (in legend) indicate a count of villages in each vulnerability/resilience category.

**Table 1 ijerph-17-01985-t001:** Major disasters that occurred in Nepal from 2008 to 2019.

	Dates	Disaster Type	Death (Missing)	Injured	Estimated Damages (NPR Million)	Impacted Districts
1	2019-03-31	Windstorm	27	1122	90	Bara, Parsa
2	2017-08-14	Flood	134	22	60,716.6 *	35 districts, 18 Tarai districts severely affected
3	2016-07-26	Landslide	25 (1)	7	-	Pyuthan
4	2015-07-30	Landslide	27 (1)	3	-	Kaski
5	2015-06-12	Landslide	39	8	-	Taplejung
6	2015-04-25	Earthquake	8970	22,302	706,461 **	31 (out of 77) districts
7	2014-10-14	Snowstorm	48	-	-	Mustang (impact of HudHud cyclone)
8	2014-08-14	Flood	34 (91)	26	10,052	Surkhet
9	2014-08-14	Flood	14 (4)	2	511.84	Dang
10	2014-08-13	Flood	33 (15)	2	3775.4	Bardiya
11	2014-08-13	Flood	15 (5)	2	480	Banke
12	2014-08-02	Landslide	33 (123)	47	130.4 ***	Sindhupalchowk
13	2014-04-18	Avalanche	13 (3)	7	-	Solukhumbu
14	2012-09-30	Landslide	10 (4)	-	-	Ilam
15	2012-09-23	Avalanche	9 (3)	13	-	Gorkha
16	2012-05-05	Flash Flood	40 (32)	5	11	Kaski
17	2011-09-18	Earthquake	6	30	-	Eastern districts
18	2008-08-18	Flood	55	2,350	3,773.6 ****	Sunsari (Koshi embankment breach)

Data source: Disaster Portal (www.drrportal.gov.np); * Post Flood Recovery Needs Assessment [14], ** Post Disaster Needs Assessment [13], *** Gaire, Delgado & González (2015) [15], **** Nepal Disaster Report (2009) [16].

**Table 2 ijerph-17-01985-t002:** Characteristics of selected community disaster resilience assessment measures.

	Index/Model (Authors)	Type	Methodological Approach	Geographic Focus (Country, Study Area)	Domains & Number of Indicators
1	Baseline Resilience Index for Communities (BRIC) (Cutter et al. 2010) [26]	Index	Disaster Resilience of Place (DROP)	United States, FEMA Region IV	Social (7), Economic (7), Institutional (8), Infrastructural (7), Community Capital (7)
2	Climate Disaster Resilience Index (CDRI) (Shaw & IDEM 2009) [33]	Index	Qualitative approach	Indonesia, Banda Aceh; Thailand, Bangkok; Sri Lanka, Colombo; Vietnam, Danang & Hue; Philippines, Iloilo & City of San Fernando; India, Mumbai; and Japan, Yokohama	Natural (2), Physical (8), Social (3), Economics (6), Institutional (4)
3	Coastal community resilience (CCR) (Courtney et al. 2008) [34]	Tool	Participatory process	Thailand, Sri Lanka, Indonesia, India, and the Maldives (Indian Ocean region)	Governance (4), Society and Economy (4), Coastal Resource Management (4), Land Use and Structural Design (4), Risk Knowledge (4), Warning and Evacuation (4), Emergency Response (4), Disaster Recovery (4)
4	Coastal Resilience Index (Sempier et al. 2010) [35]	Score card	-	USA, Gulf Coast	Community Capacities: Critical Infrastructure & Facilities, Transportation Issues, Community Plans & Agreements, Mitigation Measures, Business Plans, Social Systems
5	Communities Advancing Resilience Toolkit (CART) (Pfefferbaum et al. 2013) [36]	Tool	Qualitative, participatory approach	Individual Communities (not specified)	Connection and Caring (8), Resources (6), Transformative potential (1), Disaster Management (4)
6	Community Disaster Resilience Index (CDRI) (Yoon et al. 2015) [25]	Index	Statistical approach, factor analysis	South Korea, 229 local municipalities	Human (5), Social (3), Economic (3), Institutional (5), Physical (4), Environmental (4)
7	Community Disaster Resilience Index (Mayunga 2007) [37]	Index	Theoretical Framework Matrix	USA, Texas	Social Capital (3), Economic Capital (3), Human Capital (4), Physical Capital (3), Natural Capital (3)
8	Community Resilience Index (Kafle 2012) [27]	Index	Statistical approach	Indonesia, Aceh	Process (10), Outcome (25)
9	Community-Based Resilience Analysis (CoBRA) (UNDP 2014) [38]	Tool	Participatory qualitative approaches	Kenya and Uganda	Community Characteristics
10	Conjoint Community Resilience Assessment Measurement (CCRAM) (Cohen et al. 2013) [39]	Tool	Literature reviews and DELPHI	Israel, 9 towns	Community Capacities: Leadership, collective efficacy, preparedness, place attachment, social trust, social relationship
11	Modified BRIC (Siebeneck et al. 2015) [6]	Index	Statistical approach, factor analysis	Thailand, 76 provinces	Social (6), Economic (3), Institutional (11), Community (5)
12	PEOPLES (Reneschier et al. 2010) [40]	Tool	MCEER	USA, New York	Population and Demographics (3), Environmental/Ecosystem (6), Organized Governmental Services (3), Physical Infrastructure (2), Lifestyle & Community Competence (3), Economic Development (3), Social-Cultural Capital (7)
13	Rural Resilience Index (RRI) (Cox & Halmen 2015) [41]	Index	Participatory action research	Canada, British Columbia	Social Fabric, Community Resources, Disaster Management

**Table 3 ijerph-17-01985-t003:** List of variables used in resilience analysis.

Resilience Concept (*VARIABLE NAME*)	Variable Description	Data Source	Impact on Resilience
**Social resilience**			
Pre-retirement age (*PAGE65*)	% pop below 65 years of age	Census 2011	Positive
Transportation (*PVEHICLE*)	% households with at least one vehicle	Census 2011	Positive
Communication capacity (*PPHONE*)	% households with telephone service available	Census 2011	Positive
Information access (*PRADIO*)	% of household with access to a radio	Census 2011	Positive
Language competency (*PNEPALI*)	% pop proficient Nepali speakers	Census 2011	Positive
Non-special needs (*PNODIS*)	% pop without sensor, physical, or mental disability	Census 2011	Positive
Education (*PBSLC*)	% pop without school degree (School leaving certificate (SLC) education)	Census 2011	Negative
Female-headed households (*PFEMHH*)	% female-headed households	Census 2011	Negative
Caste (*PDALIT*)	% Dalit population	Census 2011	Negative
**Economic resilience**			
Homeownership (*POWNHH*)	% owner-occupied housing units	Census 2011	Positive
Employment rate (*PEMPLOY*)	% labor force employed	Census 2011	Positive
Non-dependence on primary sectors (*PNOAGRI*)	% pop not employed in farming, fishing, forestry, and extractive industries	Census 2011	Positive
Employment (*PFEMEMPLOY*)	% female labor force participation	Census 2011	Negative
**Community Capital**			
Place attachment (*PSAMEDIS*)	% pop born in the same place	Census 2011	Positive
5-yr migration (*PMIGRATED*)	% pop who migrated within previous 5 years	Census 2011	Negative
Absentee population (*PABSENTPOP*)	% pop who are working outside of the country	Census 2011	Negative
**Infrastructure Resilience**			
Sturdier housing types (*PRCC*)	% housing units with reinforced cement concrete (RCC) foundation	Census 2011	Positive
Internet Infrastructure (*PINTERNET*)	% of households with internet access	Census 2011	Positive
Cooking capabilities (*PHHGASELEC*)	% of households with gas and/or electric cooking capabilities	Census 2011	Positive
**Environmental resilience**			
Rainy days (*PRAINYDAY*)	Average no. of rainy days	Karki, Schickhoff [56]	Negative
Elevation (*AVGELEV*)	Average elevation	ASTER GDEM	Negative
Pervious surfaces (*PPERVIOUS*)	Average % perviousness	ICIMOD 2011	Positive

**Table 4 ijerph-17-01985-t004:** Principal components, loaded variables, factor loadings, and percentage of variance.

Component	Loaded Variables	Factor Loadings	Component Theme	% of Variance
1	PRCC	0.840	Infrastructure	26.05
	PINTERNET	0.837		
	PNOAGRI	0.703		
	PPHONE	0.539		
	PBSLC	-0.704		
	PSAMEDIS	-0.752		
	POWNHH	-0.817		
2	PFEMEMPLOY	0.840	Economic-Social	18.72
	PEMPLOY	0.796		
	PNEPALI	0.655		
	AVGRAINY	0.633		
	AVGELEV	0.591		
	PRADIO	0.506		
	PNODIS	-0.568		
	PVEHICLE	-0.748		
3	PABSENTPOP	0.805	Community Capital	8.94
	PFEMHH	0.767		
	PPHONE	0.552		
	PAGE65	-0.639		
4	AVGELEV	0.664	Environmental	5.56
	PPERVIOUS	-0.789		
5	PDALIT	0.842	Caste	5.01
6	PMIGRATED	0.989	Migration	4.77
Total				**69.05**

**Table 5 ijerph-17-01985-t005:** Distribution of village development committees (VDCs) and municipalities in each CDRI categories across the ecoregion.

Eco-Region	Total Number of Districts	Total Number of VDCs	Number (and %) of VDCs in Each CDRI Categories				
			High	High-Medium	Medium	Medium-Low	Low
Mountain	16	544	31 (5.7%)	114 (20.9%)	319 (58.6%)	79 (14.5%%)	1 (0.2%)
Hill	39	2033	200 (9.8%)	603 (29.7%)	937 (46.1%)	290 (14.3%)	3 (0.1%)
Tarai	20	1394	7 (0.5%)	87 (6.2%)	459 (32.9%)	703 (50.4%)	138 (9.9%)

**Table 6 ijerph-17-01985-t006:** Summary of VDCs and population in each CDRI class by the district.

	District	Total VDCs	% of VDCs in each category of DRI					Total Population Census 2011	% population in each category of DRI				
			High	High-Medium	Medium	Medium-Low	Low		High	High-Medium	Medium	Medium-Low	Low
1	Taplejung	50	6.00%	24.00%	68.00%	2.00%	-	126,448	2.51%	35.14%	60.74%	1.61%	-
2	Panchthar	41	-	9.76%	90.24%	-	-	190,491	-	16.30%	83.70%	-	-
3	Ilam	49	-	4.08%	77.55%	18.37%	-	287,932	-	7.73%	74.74%	17.53%	-
4	Jhapa	50	4.00%	14.00%	50.00%	32.00%	-	807,934	7.53%	28.25%	47.09%	17.12%	-
5	Morang	66	1.52%	12.12%	46.97%	37.88%	1.52%	959,568	0.59%	15.53%	59.05%	23.98%	0.85%
6	Sunsari	52	3.85%	3.85%	50.00%	40.38%	1.92%	753,244	18.80%	13.67%	41.37%	24.96%	1.20%
7	Dhankuta	36	-	5.56%	83.33%	11.11%	-	161,398	-	19.36%	67.95%	12.69%	-
8	Terhathum	32	-	21.88%	71.88%	6.25%	-	100,833	-	24.68%	70.12%	5.19%	-
9	Sankhuwasabha	34	5.88%	2.94%	70.59%	17.65%	2.94%	158,222	7.05%	16.62%	64.38%	11.72%	0.23%
10	Bhojpur	63	-	6.35%	74.60%	19.05%	-	181,225	-	9.30%	72.13%	18.57%	-
11	Solukhumbu	34	5.88%	23.53%	58.82%	11.76%	-	105,119	5.35%	21.56%	58.68%	14.41%	-
12	Khotang	76	1.32%	6.58%	75.00%	17.11%	-	205,225	0.81%	4.65%	77.70%	16.84%	-
13	Okhaldhunga	56	-	3.57%	76.79%	19.64%	-	146,824	-	5.22%	75.33%	19.44%	-
14	Udayapur	45	-	-	55.56%	44.44%	-	315,429	-	-	76.84%	23.16%	-
15	Saptari	115	-	0.87%	53.04%	46.09%	-	637,844	-	0.96%	56.47%	42.57%	-
16	Siraha	108	-	4.63%	57.41%	37.96%	-	635,627	-	6.61%	58.73%	34.66%	-
17	Dhanusa	102	-	9.80%	54.90%	34.31%	0.98%	753,682	-	21.39%	47.93%	29.75%	0.93%
18	Mahottari	77	-	1.30%	35.06%	61.04%	2.60%	625,207	-	1.07%	35.23%	61.45%	2.25%
19	Sarlahi	100	-	1.00%	5.00%	72.00%	22.00%	769,330	-	1.33%	7.73%	73.99%	16.95%
20	Sindhuli	54	-	1.85%	42.59%	55.56%	-	293,173	-	13.44%	37.11%	49.45%	-
21	Ramechhap	55	-	5.45%	76.36%	18.18%	-	201,423	-	7.67%	72.67%	19.66%	-
22	Dolakha	52	5.77%	15.38%	71.15%	7.69%	-	185,099	5.45%	23.39%	65.70%	5.46%	-
23	Sindhupalchok	79	2.53%	11.39%	69.62%	16.46%	-	285,770	3.45%	13.96%	66.30%	16.29%	-
24	Kavrepalanchok	90	1.11%	3.33%	45.56%	50.00%	-	375,221	6.60%	5.98%	46.39%	41.03%	-
25	Lalitpur	42	23.81%	16.67%	35.71%	23.81%	-	457,606	72.04%	13.19%	10.35%	4.42%	-
26	Bhaktapur	18	44.44%	16.67%	38.89%	0.00%	-	298,704	77.82%	7.01%	15.17%	-	-
27	Kathmandu	59	54.24%	16.95%	27.12%	1.69%	-	1,699,289	92.52%	3.45%	3.77%	0.27%	-
28	Nuwakot	62	-	4.84%	43.55%	50.00%	1.61%	275,775	-	11.75%	42.33%	45.02%	0.90%
29	Rasuwa	18	-	22.22%	72.22%	5.56%	-	42,133	-	17.42%	77.43%	5.15%	-
30	Dhading	50	2.00%	12.00%	64.00%	22.00%	-	334,292	6.04%	11.07%	61.15%	21.75%	-
31	Makwanpur	44	2.27%	-	25.00%	68.18%	4.55%	415,601	20.37%	-	27.77%	49.56%	2.29%
32	Rautahat	97	-	-	2.06%	62.89%	35.05%	686,059	-	-	0.58%	68.95%	30.47%
33	Bara	99	-	-	7.07%	60.61%	32.32%	685,831	-	-	8.14%	65.55%	26.30%
34	Parsa	83	-	1.20%	7.23%	68.67%	22.89%	597,769	-	0.58%	26.87%	55.08%	17.47%
35	Chitawan	38	2.63%	39.47%	36.84%	13.16%	7.89%	569,732	25.25%	38.75%	27.92%	5.55%	2.53%
36	Gorkha	67	8.96%	31.34%	47.76%	11.94%	-	268,942	19.63%	28.50%	45.76%	6.10%	-
37	Lamjung	61	24.59%	50.82%	21.31%	3.28%	-	166,150	36.90%	43.65%	17.12%	2.33%	-
38	Manang	13	-	15.38%	61.54%	15.38%	-	5,553	-	27.12%	65.08%	7.80%	-
39	Kaski	45	26.67%	64.44%	8.89%	-	-	480,952	74.60%	23.48%	1.92%	-	-
40	Tanahu	47	34.04%	48.94%	17.02%	-	-	320,547	45.50%	42.99%	11.51%	-	-
41	Syangja	62	12.90%	59.68%	27.42%	-	-	288,100	16.96%	61.62%	21.42%	-	-
42	Parbat	55	25.45%	63.64%	10.91%	-	-	145,667	32.26%	59.60%	8.14%	-	-
43	Baglung	60	50.00%	50.00%	-	-	-	266,630	53.56%	46.44%	-	-	-
44	Myagdi	41	34.15%	58.54%	4.88%	-	-	109,606	43.02%	54.64%	2.34%	-	-
45	Mustang	16	12.50%	37.50%	31.25%	18.75%	-	11,593	12.40%	52.34%	19.54%	15.72%	-
46	Palpa	66	1.52%	37.88%	53.03%	7.58%	-	258,893	11.24%	37.62%	44.01%	7.13%	-
47	Nawalparasi	74	-	14.86%	43.24%	41.89%	-	638,954	-	25.72%	45.52%	28.75%	-
48	Rupandehi	71	1.41%	11.27%	16.90%	61.97%	8.45%	874,566	13.55%	24.08%	15.82%	39.63%	6.93%
49	Kapilbastu	78	-	2.56%	26.92%	67.95%	2.56%	569,834	-	3.54%	32.91%	61.42%	2.12%
50	Arghakhanchi	42	9.52%	73.81%	16.67%	-	-	196,895	13.29%	66.20%	20.51%	-	-
51	Gulmi	79	5.06%	81.01%	13.92%	-	-	279,005	8.14%	78.43%	13.43%	-	-
52	Rukum	43	4.65%	20.93%	69.77%	4.65%	-	207,290	2.19%	20.95%	73.62%	3.24%	-
53	Salyan	47	-	6.38%	74.47%	19.15%	-	241,716	-	7.30%	73.55%	19.15%	-
54	Rolpa	51	-	29.41%	68.63%	1.96%	-	221,177	-	28.71%	69.26%	2.02%	-
55	Pyuthan	49	18.37%	67.35%	14.29%	-	-	226,796	20.86%	67.06%	12.08%	-	-
56	Dang	41	-	17.07%	48.78%	34.15%	-	548,141	-	25.38%	42.19%	32.42%	-
57	Banke	47	-	8.51%	29.79%	44.68%	17.02%	485,164	-	21.17%	35.72%	30.42%	12.69%
58	Bardiya	32	-	-	28.13%	68.75%	3.13%	423,611	-	-	28.79%	69.20%	2.01%
59	Surkhet	51	5.88%	37.25%	47.06%	9.80%	-	343,318	7.57%	42.73%	45.49%	4.21%	-
60	Jajarkot	30	3.33%	30.00%	60.00%	6.67%	-	170,106	1.43%	24.58%	69.79%	4.21%	-
61	Dailekh	56	-	32.14%	62.50%	5.36%	-	260,855	-	31.12%	64.54%	4.35%	-
62	Dolpa	23	21.74%	43.48%	17.39%	17.39%	-	36,128	20.03%	52.18%	14.09%	13.70%	-
63	Jumla	30	13.33%	30.00%	36.67%	20.00%	-	107,495	9.92%	36.33%	35.06%	18.69%	-
64	Kalikot	30	-	23.33%	66.67%	10.00%	-	136,587	-	26.12%	66.49%	7.39%	-
65	Mugu	24	8.33%	12.50%	58.33%	20.83%	-	54,832	4.48%	14.81%	69.40%	11.32%	-
66	Humla	27	7.41%	18.52%	55.56%	18.52%	-	49,933	11.83%	24.19%	50.00%	13.99%	-
67	Bajhang	47	4.26%	27.66%	48.94%	19.15%	-	194,701	3.54%	25.83%	47.54%	23.08%	-
68	Bajura	27	7.41%	40.74%	44.44%	7.41%	-	134,154	9.43%	40.66%	44.57%	5.34%	-
69	Achham	75	5.33%	60.00%	33.33%	1.33%	-	256,188	2.84%	61.90%	33.44%	1.82%	-
70	Doti	51	5.88%	50.98%	41.18%	1.96%	-	207,070	6.81%	54.35%	37.37%	1.47%	-
71	Kailali	44	-	4.55%	38.64%	45.45%	11.36%	766,659	-	7.44%	54.05%	32.10%	6.41%
72	Kanchanpur	20	-	10.00%	60.00%	25.00%	5.00%	448,503	-	7.61%	73.72%	17.06%	1.62%
73	Dadeldhura	21	-	33.33%	57.14%	9.52%	-	141,004	-	32.29%	60.83%	6.88%	-
74	Baitadi	63	-	11.11%	73.02%	15.87%	-	250,225	-	15.63%	66.68%	17.70%	-
75	Darchula	41	-	14.63%	58.54%	26.83%	-	132,484	-	17.63%	55.41%	26.96%	-

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
