# Peer review of "Benchmarking Community Disaster Resilience in Nepal"

_ijerph, 2020, doi:10.3390/ijerph17061985_

Round 1

Reviewer 1 Report

First, I wanted to thank the authors for this interesting, well-written, and helpful study. I believe that you did a great job and it’s very well presented. However, I have a few suggestions that can potentially improve your publication.

  1. Just a minor comment for Line 95: double verb, missing closing quotes.
  2. In the introduction section, it would be helpful to have a table with a list of major hazards and disasters that occurred in Nepal in the past 10 years or so, and provide info in the table regarding: date, location in Nepal, type of disaster (i.e. earthquake, flood, fire, etc), casualties rate (i.e. number of people killed or injured related to total population) and estimated damage in dollars or other. This will provide the reader with a better understanding of the situation in Nepal, the importance of your study, and the need for a resilience framework. A similar table was used in “A GIS-based approach to identify the spatial variability of social vulnerability to seismic hazard in Italy” by Frigerio et al. (2016): “Table 1. Major natural disasters occurred in Italy from 2000 to 2013.”
  3. In section 2.2: Measuring resilience, I understand that the authors opted for the DROP model. However, other approaches exist in the literature that were not mentioned by the authors, such as “PEOPLES” resilience framework, Community Resilience Index (CRI), or Conjoint Community Resiliency Assessment Measure (CCRAM). It would be helpful to mention approaches other than DROP, briefly explain in what they differ/resemble DROP, and why DROP is the preferred approach. As an example, you could look at “What Do We Mean by ‘Community Resilience’? A Systematic Literature Review of How It Is Defined in the Literature” by Patel S. S. et al. (2017). In this article, “Table 3. Comparison between Elements Found in Review to Widely Cited Reviews, Models and Measurements” provides a good overview and makes it possible to visualize the similarities/differences among methods.
  4. Line 152: “The DROP model presents a comprehensive framework for evaluating overall community disaster resilience and provides an opportunity to pinpoint those specific components that may enhance or reduce disaster resilience. Furthermore, the application of the DROP model permits an understanding of geographic patterns of disaster resilience and enables spatial comparison among communities.” I believe that a list of elements considered by DROP such as ‘training and education’, ‘mental health’, and ‘public support’ would give a better understanding on what are the factors that should be included in the analysis. I understand that the authors offered a list of variables in “Table 1: List of variables used in resilience analysis” and I suggest to provide with a table that show a more generic list of variables and highlight which variables the authors included in their study, justifying their decisions for those they did not include.
  5. Are there regions in Nepal where it is difficult to gather data? Or data is for any reason not reliable or statistically not significant (i.e. sample size is considerably too small because of small population when compared to regions with way higher population). In this case, I would suggest to include a map with the regions where data are more available and statistically significant. This way, it will be easier for the reader to understand where the results are more reliable, and where there is the need of gathering more data for future studies.

Thank you for your excellent work. I look forward to have a look at your revised version. 

Author Response

Dear Reviewers:

Thank you very much for the insightful commentary. We appreciate your valuable comments, which have greatly assisted in the improvement of our manuscript. Please find below a detailed account of how we have addressed each of your specific concerns. All the changes in the document, relevant references, and responses are colored in Red.

Sincerely,

Authors

Reviewer 1

First, I wanted to thank the authors for this interesting, well-written, and helpful study. I believe that you did a great job and it's very well presented. However, I have a few suggestions that can potentially improve your publication.

  • Just a minor comment for Line 95: double verb, missing closing quotes.

Response: Thank you so much for your observation. We have corrected with quotation marks.

  • In the introduction section, it would be helpful to have a table with a list of major hazards and disasters that occurred in Nepal in the past 10 years or so, and provide info in the table regarding: date, location in Nepal, type of disaster (i.e. earthquake, flood, fire, etc), casualties rate (i.e. number of people killed or injured related to total population) and estimated damage in dollars or other. This will provide the reader with a better understanding of the situation in Nepal, the importance of your study, and the need for a resilience framework. A similar table was used in "A GIS-based approach to identify the spatial variability of social vulnerability to seismic hazard in Italy" by Frigerio et al. (2016): "Table 1. Major natural disasters occurred in Italy from 2000 to 2013."

Response: Thank you so much for your suggestion. We have added new "Table 1: Major disasters occurred in Nepal from 2008 to 2019" on page 2, line 86. The table provides information on the date of disastrous events, deaths and missing, injured, estimated damages, and impacted districts. We also added a sentence on page 2, line 77-78, to reflect the addition of the table.

  • In section 2.2: Measuring resilience, I understand that the authors opted for the DROP model. However, other approaches exist in the literature that were not mentioned by the authors, such as "PEOPLES" resilience framework, Community Resilience Index (CRI), or Conjoint Community Resiliency Assessment Measure (CCRAM). It would be helpful to mention approaches other than DROP, briefly explain in what they differ/resemble DROP, and why DROP is the preferred approach. As an example, you could look at "What Do We Mean by 'Community Resilience'? A Systematic Literature Review of How It Is Defined in the Literature" by Patel S. S. et al. (2017). In this article, "Table 3. Comparison between Elements Found in Review to Widely Cited Reviews, Models and Measurements" provides a good overview and makes it possible to visualize the similarities/differences among methods.

Response: Thanks a lot for the suggestion. We have now added "Table 2: Characteristics of selected community disaster resilience assessment measures" on page 4, line 155. The table consists of information on selected community resilience assessment methods, its type, domains and number of indicators, methodological approach, and geographic focus. We also added a few sentences on page 4, lines 145-153, to highlight the literature on a systematic review of community disaster resilience and justify our selection of the BRIC model.

  • Line 152: "The DROP model presents a comprehensive framework for evaluating overall community disaster resilience and provides an opportunity to pinpoint those specific components that may enhance or reduce disaster resilience. Furthermore, the application of the DROP model permits an understanding of geographic patterns of disaster resilience and enables spatial comparison among communities." I believe that a list of elements considered by DROP such as 'training and education', 'mental health', and 'public support' would give a better understanding on what are the factors that should be included in the analysis. I understand that the authors offered a list of variables in "Table 1: List of variables used in resilience analysis" and I suggest to provide with a table that show a more generic list of variables and highlight which variables the authors included in their study, justifying their decisions for those they did not include.

Response: Thank you so much for your valuable comments. We have added a few sentences in 'selection of variables' section (page 8, lines 231-242), providing details on the DROP model and our reasonings on variables selection. We have made similar points in the discussion section (page 16, lines 456-459) of the manuscript as well.

  • Are there regions in Nepal where it is difficult to gather data? Or data is for any reason not reliable or statistically not significant (i.e. sample size is considerably too small because of small population when compared to regions with way higher population). In this case, I would suggest to include a map with the regions where data are more available and statistically significant. This way, it will be easier for the reader to understand where the results are more reliable, and where there is the need of gathering more data for future studies.

Response: Thank you so much for the questions. We have added a few sentences at the end of the discussion section (page 18, line 466-472) to highlight the challenges of data gatherings.

Reviewer 2 Report

REVIEW - "Benchmarking community disaster resilience in Nepal"

The article is well structured and has a good methodology sector. The fact of the research has well done sectors brings a good discussion part (that is a logical affirmation). In other words, it is a good article and the reviewer has three considerations:

1) Figure 1: This is an interesting administrative map of Nepal but the authors need to insert the geographical coordinates within it.

2) Table 3: Despite of the relevance of the results, I think that the table could be better organized at its layout. It is a bit messy.

3) Conclusions: I think that this is a very good research to have a so poor conclusion. Improve it!

Thank you.

Author Response

Dear Reviewers:

Thank you very much for the insightful commentary. We appreciate your valuable comments, which have greatly assisted in the improvement of our manuscript. Please find below a detailed account of how we have addressed each of your specific concerns. All the changes in the document, relevant references, and responses are colored in Red.

Sincerely,

Authors

Reviewer 2:

REVIEW - "Benchmarking community disaster resilience in Nepal"

The article is well structured and has a good methodology sector. The fact of the research has well done sectors brings a good discussion part (that is a logical affirmation). In other words, it is a good article and the Reviewer has three considerations:

1) Figure 1: This is an interesting administrative map of Nepal but the authors need to insert the geographical coordinates within it.

Response: Thank you so much for your suggestion. Now, we have added a new figure, "Figure 1: Administrative map of Nepal" with coordinates (line 195).

2) Table 3: Despite of the relevance of the results, I think that the table could be better organized at its layout. It is a bit messy.

Response: Thank you so much for your concern. It is because of the manuscript template as it asks us to submit all the tables within the manuscript and in portrait form. We believe it will be taken care of during the production phase of the manuscript if we were accepted for the publication.

3) Conclusions: I think that this is a very good research to have a so poor conclusion. Improve it!

Response: Thank you so much for your valuable insights. We have edited and added more sentences to make the conclusion clearer and stronger (page 21, lines 480-486, 490-494, 498-504).

Round 2

Reviewer 1 Report

Thank you for improving your paper according to the comments provided. I believe that, especially nowadays, this research topic is highly valuable.

All the best.